# Characterization of hFOB 1.19 Cell Line for Studying Zn-Based Degradable Metallic Biomaterials

**DOI:** 10.3390/ma17040915

**Published:** 2024-02-16

**Authors:** Eva Jablonská, Lucie Mrázková, Jiří Kubásek, Dalibor Vojtěch, Irena Paulin, Tomáš Ruml, Jan Lipov

**Affiliations:** 1Department of Biochemistry and Microbiology, University of Chemistry and Technology, Prague, Technická 5, 166 28 Prague 6, Czech Republic; luciemrazkova3@gmail.com (L.M.); rumlt@vscht.cz (T.R.); lipovj@vscht.cz (J.L.); 2Department of Metals and Corrosion Engineering, University of Chemistry and Technology, Prague, Technická 5, 166 28 Prague 6, Czech Republic; kubasekj@vscht.cz (J.K.); vojtechd@vscht.cz (D.V.); 3Institute of Metals and Technology, Ljubljana, Lepi pot 11, SI-1000 Ljubljana, Slovenia; irena.paulin@imt.si

**Keywords:** zinc degradable materials, *in vitro* cytotoxicity testing, hFOB 1.19 osteoblasts

## Abstract

*In vitro* testing is the first important step in the development of new biomaterials. The human fetal osteoblast cell line hFOB 1.19 is a very promising cell model; however, there are vast discrepancies in cultivation protocols, especially in the cultivation temperature and the presence of the selection reagent, geneticin (G418). We intended to use hFOB 1.19 for the testing of Zn-based degradable metallic materials. However, the sensitivity of hFOB 1.19 to zinc ions has not yet been studied. Therefore, we compared the toxicity of zinc towards hFOB 1.19 under different conditions and compared it with that of the L929 mouse fibroblast cell line. We also tested the cytotoxicity of three types of Zn-based biomaterials in two types of media. The presence of G418 used as a selection reagent decreased the sensitivity of hFOB 1.19 to Zn^2+^. hFOB 1.19 cell line was more sensitive to Zn^2+^ at elevated (restrictive) temperatures. hFOB 1.19 cell line was less sensitive to Zn^2+^ than L929 cell line (both as ZnCl_2_ and extracts of alloys). Therefore, the appropriate cultivation conditions of hFOB 1.19 during biomaterial testing should be chosen with caution.

## 1. Introduction

The first important step in the development of any biomaterial is *in vitro* cytotoxicity testing, although it cannot encompass all of the complexities in the body. In addition to biomaterials intended for permanent bone implants, there is also an increasing interest in degradable metallic biomaterials for temporary bone fixation (reviewed in [1]). Here, we focus on zinc-based biomaterials for orthopedic applications, which have been the subject of interest in the last decade [2,3,4,5]. The corrosion rate and toxicity of *in vitro* tests often do not correlate with those *in vivo* (e.g., [5,6,7]). There are probably many reasons for this (e.g., lack of circulation of body fluids, only one cell type used, 2D cell arrangement *in vitro* [8,9,10]); nevertheless, the appropriate cell model can also play a prominent role [11].

There are two main options for choosing the *in vitro* model: primary cells or cancer-derived cell lines. The former reliably mimics properties of the tissue *in vivo*; however, the cells have a limited lifespan and a high variability within donors, which hinders their routine use in experiments. Continuous cell lines represent an unlimited cell source for testing, since they can proliferate indefinitely. However, their behavior differs significantly from that of primary cells, and thus of the cells in tissues. A compromise choice may be cells obtained by so-called gentle cell immortalization, for example, their transfection with a gene coding a temperature-sensitive mutant of the SV40 large T antigen (tsA58), since it can lead to a combination of advantageous properties of both types mentioned above. At a lower (permissive) temperature, tsA58 is active and promotes cell proliferation. At elevated (restrictive) temperature, cells stop dividing and can differentiate into mature cell types [12,13].

The ISO 10993-5 standard [14] mentions the L929 murine fibroblast cell line (ATCC CCL-1) as a possible cell model to be used in *in vitro* biomaterial testing. Specifically, for the cytocompatibility testing of biomaterials intended for orthopedic applications, osteoblast-like cancer cell lines (e.g., U-2 OS, MG-63, and Saos-2) are used; however, none of them can be considered a reliable substitute for human osteoblasts [15,16].

To date, the biological response of newly prepared Zn alloys has been performed on cell cultures of various origins (reviewed, e.g., in [17]), mainly on L929 and on osteoblast-like cell lines MG-63 and MC3T3-E1 [18]. However, there is a demand to use an appropriate and preferably non-cancerous cell line for the intended application for cytotoxicity evaluation [11]. We have previously shown that the cancerous osteoblast-like cell line U-2 OS is less sensitive than the L929 cell line to zinc ions [19]. Here, we present the conditionally immortalized human fetal osteoblast cell line hFOB 1.19 (ATCC CRL-11372), established by Harris et al. [20], as a suitable model for degradable zinc-based biomaterials, as it is not cancerous and has the characteristics of osteoblasts.

The cell line hFOB 1.19 was derived from biopsies of limb tissues from a spontaneous miscarriage and immortalized by transfection with tsA58. The selection reagent, G418 (geneticin), facilitated the initial selection of transfected cells and, at a reduced concentration (300 μg∙mL^−1^), it ensures further maintenance of the transgene cells during long-term cultivation [20].

The cell line hFOB 1.19 was further characterized by Subramaniam et al., who showed that this cell line was capable of bone formation *in vivo*, and the analysis of the karyotype showed only minimal chromosomal changes [21]. Furthermore, Yen et al. observed the multilineal differentiation potential (toward adipocytes or chondrocytes) of hFOB 1.19 cells under appropriate conditions [22]. Marozin et al. even claimed that hFOB 1.19 can serve as a surrogate model for bone marrow-derived mesenchymal stromal cells [23].

However, there are also great discrepancies in the cultivation protocols. There is disagreement on the interpretation of the restrictive temperature. ATCC, the cell supplier, states 39.5 °C in their protocols, but some authors choose 37 °C for their experiments for culture differentiation [24,25,26,27]. On the contrary, some authors consider 37 °C to be a permissive temperature [28,29]. It is also not clear whether the G418 selection reagent should be added during osteodifferentiation.

The hFOB 1.19 cell line has already been used in cancer research and osteomyelitis research [25,30] as well as in studies with permanent [31,32] and degradable [24,33] biomaterials. There are also studies that use hFOB 1.19 for testing degradable materials doped with Zn (Table 1). However, the sensitivity of the hFOB 1.19 cell line to Zn^2+^ has not yet been studied.

Here, we compare the sensitivity of the L929 and hFOB 1.19 cell lines to Zn^2+^ in different media, as well as the effect of temperature on the metabolic activity of hFOB 1.19 and on its sensitivity to Zn^2+^.

## 2. Materials and Methods

### 2.1. Cell Cultivation

hFOB 1.19 cells (ATCC, Manassas, VA, USA, CRL-1137) were maintained in DMEM/Ham’s F-12 medium without phenol red (Sigma–Merck, Darmstadt, Germany, D6434) supplemented with 10% FBS (Sigma, F7524), 2.5 mM of L-glutamine (Sigma, G7513), and the selection reagent, G418 (Sigma, G8168), at a final concentration of 0.3 mg·mL^−1^ at 34 °C, 5% CO_2_, and 100% relative humidity. Cells were passaged regularly when sub-confluent using a trypsin-EDTA solution without phenol red (Gibco, 15400054, Thermo Fisher Scientific, Waltham, MA, USA).

Murine fibroblasts L929 (ATCC CCL-1) were maintained in an MEM (Sigma, M0446) medium supplemented with 10% FBS (FBS, Sigma F7524) under standard conditions of 37 °C, 5% CO_2_ and 100% relative humidity. Cells were passaged regularly when sub-confluent using trypsin-EDTA solution (Sigma, T4049).

### 2.2. Preparation of the ZnCl_2_ Solution

The 10 mM stock solution of ZnCl_2_ was prepared in dH_2_O and sterilized by filtration. Fresh working solutions (from 40 µM up to 360 µM) were prepared in a cultivation medium prior to the experiment.

### 2.3. Preparation of the Materials

In the present study, three different materials, which were considered promising for future applications in the development of medical devices (Table 2), were used for cytotoxicity tests. The materials were prepared similarly as in [40,41,42,43]. Zn-1Mg and Zn-1Mg-1Ag (wt.%) were prepared by mechanical alloying using pure zinc powder (99.9%, particle size < 149 µm, Thermo Fisher Scientific), pure magnesium (99.8%, particle size < 44 µm, Alfa Aesar-Thermo Fisher Scientific), and pure Ag (99.5%, particle size < 44 µm, Safina a.s., Vestec, Czech Republic) as the input materials. To prevent the agglomeration of powder particles during milling, 0.03 g of stearic acid was added to all selected compositions. The mechanical alloying was performed for 4 h in ZrO2 vessels under an argon-protective atmosphere (purity 99.95%) using a Retsch E-Max milling machine with 800 rotations per minute (RPM). The milling balls were composed of ZrO2, while the ball-to-powder weight ratio was equal to 5:1. The weight of the input powder mixture was 30 g. The mill equipped with a water-cooling system enabled us to keep the temperature of the process low. The temperature of the cooling medium was maintained between 30 and 50 °C. Zn-0.8Mg-0.2Ag was prepared as follows: Commercially available zinc (99.995 wt%), magnesium (Magnesium Elektron, Manchester, UK, 99.95 wt%), and silver (Safina, a.s., 99.0 wt%) were used for the preparation of the alloys by the general casting process. The Zn-0.8Mg-0.2 Ag (wt%) alloy was melted in an electrical resistance furnace in a graphite crucible. Pure Zn was melted at 520 °C; subsequently, pure Mg and Ag were added and the melt was manually stirred using a graphitic rod. It was subsequently left at 520 °C for 10 min in a resistance furnace. No protective atmosphere was used. Finally, the melt was poured into a non-preheated steel cylindrical mold with a diameter of 50 mm and length of 400 mm. To homogenize the microstructure, the obtained ingots were thermally treated at 350 °C for 24 h in air with subsequent quenching into water (25 °C). Subsequently, billets for the extrusion process, 30 mm in diameter and 35 mm in length, were machined. The surface of these billets was finished by conventional mechanical machining. The billets were extruded at 200 °C using extrusion dies with a die angle of 90° and an extrusion ratio (ER) of 25:1. The billets were tempered for 10 min in the extrusion die. Subsequently, the extrusion was performed at a constant ram speed of 0.2 mm/s and the final rod was cooled in air.

Cylindrical samples (5 mm in diameter and 2 mm in height) were ground (SiC paper, with a grit size up to P4000), cleaned, and sterilized by immersion in 70% ethanol (2 h) and by subsequent exposure to UV (2 h).

### 2.4. Preparation of the Extracts

The prepared samples were transferred to MEM or DMEM/Ham’s F-12 cultivation medium with 5% fetal bovine serum (FBS) and agitated (130 RPM) at 37 °C in closed vessels for 24 h. The surface-to-volume ratio was 87.5 mm^2^·mL^−1^ for all samples. Four replicates were used for each sample for each medium. The extracts from four samples were then pooled. The extracts were used for indirect *in vitro* cytotoxicity tests (undiluted, i.e., 100%, and diluted., i.e., 50% extracts were used) and for ICP-MS measurement.

### 2.5. Metabolic Activity of hFOB 1.19 and Cytotoxicity of Zn^2+^ towards hFOB 1.19 after Long-Term Cultivation at Different Temperatures

hFOB 1.19 cells were seeded in 48-well plates in DMEM/Ham’s F-12 medium without phenol red, with 10% FBS and with or without the selection reagent, G418. Forty-eight-well plates were used to facilitate medium exchanges. The seeding density was 263,000 cells∙cm^−2^. The cells were cultivated at 34 °C for 3 days until full confluence was reached. Subsequently, the cultivation continued at different temperatures (34, 37, or 39.5 °C). The medium was exchanged every 4–5 days. After 14 days of cultivation, the medium was exchanged for solutions of ZnCl_2_ in the medium with 5% FBS. After 24 h, the metabolic activity was evaluated using the resazurin assay.

### 2.6. Cytotoxicity of Zn^2+^ and Extracts towards hFOB 1.19 and L929 in Different Media

hFOB 1.19 or L929 cells were seeded in 96-well plates (20,000 cells/well in 50 µL, i.e., approx. 60,000 cells∙cm^−2^) in DMEM/Ham’s F-12 or in the MEM medium with 10% FBS and cultivated for 4 h at 34 °C or 37 °C, respectively. Subsequently, ZnCl_2_ solutions in media without FBS (50 µL) were added to the cells to achieve the 5% FBS concentration. After one day, the metabolic activity was evaluated using the resazurin assay.

### 2.7. Zn^2+^ and Extracts’ Cytotoxicity towards hFOB 1.19 and L929 in Different Media

hFOB 1.19 or L929 cells were seeded in 96-well plates (20,000 cells/well in 100 µL, i.e., approx. 60,000 cells∙cm^−2^) in DMEM/Ham’s F-12 or in the MEM medium with 10% FBS and cultivated for 4 h at 34 °C or 37 °C, respectively. Subsequently, the medium was replaced with alloy extracts in medium with 5% FBS (100 µL). After one day, the metabolic activity was evaluated using the resazurin assay.

### 2.8. Evaluation of Metabolic Activity (Resazurin Assay)

Cell metabolic activity was evaluated using the resazurin assay [44]. Resazurin is metabolized to resorufin by living cells. The cell medium was aspirated and a resazurin solution (final concentration: 25 µg∙mL^−1^) in a medium without phenol red was added. After 1 h of incubation, the fluorescence was measured at 560/590 nm (excitation/emission) (Fluoroskan Ascent FL, Thermo, or Spectramax id5, Molecular Devices). Metabolic activity was depicted as the absolute fluorescence of resorufin. The cytotoxicity of the solutions and extracts was depicted as a percentage of the metabolic activity of the control (sole medium without zinc). Extracts that caused a decrease below 70% of the activity of the control were considered cytotoxic, as described in the ISO 10993-5 standard [14].

### 2.9. ICP-MS Measurement

The extracts were analyzed by inductively coupled plasma–mass spectrometry (ICP-MS) using a Perkin Elmer Elan 6000 spectrometer (three measurements for each sample). Prior to the measurement, ultrapure HNO_3_ was added to the extracts in order to dissolve the majority of solid corrosion products.

### 2.10. Statistical Analysis

A one-way ANOVA followed by Tukey’s test was performed in the R software, version 4.2.3. Significant differences (*p* < 0.05) among groups are indicated by letters.

## 3. Results

Our main goal was to determine the sensitivity of hFOB 1.19 cells to Zn^2+^ at both permissive and restrictive temperatures. Due to the discrepancies in the published cultivation protocols, we used a temperature of 34 °C as proliferative and both temperatures of 37 °C and 39.5 °C as restrictive. We also used media with and without the G418 selection reagent. The effect of zinc ions on cells under different conditions was then evaluated. We compared the results obtained for hFOB 1.19 with those of the L929 cell line, which is recommended for cytotoxicity tests according to the ISO-10993 standard [14]. We also used Zn-based biomaterials and performed a test with the extract according to ISO 10993-5 [14] using both cell types.

### 3.1. The Influence of Selection Reagent on the Metabolic Activity of hFOB 1.19

We compared the metabolic activity of cells after 7 and 14 days of incubation at different temperatures with and without the G418 selection reagent (Figure 1). After 7 days, the metabolic activity was significantly lower in the absence of G418 at both restrictive temperatures compared to the permissive one. After 14 days, the difference between the variants with and without G418 was less profound and, surprisingly, was statistically significant both at the restrictive temperature and the permissive temperatures of 39.5 °C and 34 °C, respectively. 

### 3.2. Zn^2+^ Was More Toxic to hFOB 1.19 Cells at Restrictive Temperatures Than at Permissive Temperatures

Figure 2 shows the metabolic activity (resazurin assay) of hFOB 1.19 cells after their one-day exposure to ZnCl_2_ solutions preceded by 14-day incubation at different temperatures. No toxic effect of zinc ions was observed in cells cultured at 34 °C, even at the highest concentration of Zn^2+^ used (140 μmol∙L^−1^). Cells cultured in the absence of G418 at restrictive temperatures were more sensitive to ZnCl_2_ compared to those cultured in the presence of G418. The decrease below the normative cut-off was observed at a concentration of 140 μmol∙L^−1^.

The possibility of G418 influencing the metabolic activity assay was excluded, because the medium with G418 was removed prior to the addition of ZnCl_2_ solutions.

### 3.3. Zn^2+^ Was More Toxic to L929 Than to hFOB 1.19 in the Media Recommended for Each Cell Line

To compare the toxic effect of Zn^2+^ on L929 and hFOB 1.19, the same numbers of cells were seeded, and after 4 h of incubation, the ZnCl_2_ solutions were added.

Figure 3 shows that the mouse fibroblast cell line L929 was more sensitive to Zn^2+^ than hFOB 1.19 cells (a decrease in metabolic activity below 70% was observed at a 60 μmol∙L^−1^ concentration of ZnCl_2_, while a toxic effect towards hFOB 1.19 cells was observed at 100 μmol∙L^−1^ of ZnCl_2_).

### 3.4. The Effects of Media Used

Different cultivation media for the cell lines were used in the previous experiment according to the recommendation of the manufacturers. To evaluate the effect of the cultivation medium on the toxicity of Zn^2+^, a test was performed with both MEM and DMEM/Ham’s F-12 culture medium used for both the hFOB 1.19 and L929 cell lines. Again, ZnCl_2_ solutions were added 4 h after seeding the same number of both cell lines. We did not use G418 in this experiment to ensure the same conditions for both cell lines.

As in the previous experiment (Figure 3), the higher sensitivity of L929 compared to hFOB 1.19 cells was shown and the toxic effect at the 60 μmol∙L^−1^ concentration of Zn^2+^ was also confirmed for the L929 cell line (Figure 4). A higher metabolic activity of the L929 cells was observed in the MEM medium, which is recommended for this cell line. On the contrary, the hFOB 1.19 cells withstood higher concentrations of ZnCl_2_ in the DMEM/Ham’s F-12 medium, which is recommended for this cell line. The threshold toxic concentration of ZnCl_2_ in the case of hFOB 1.19 cells in MEM was around 100 μmol∙L^−1^.

### 3.5. Comparison of the Cytotoxicity of Extracts of Alloys in Different Media towards hFOB 1.19 and L929

We also tested extracts of selected Zn-based degradable biomaterials according to ISO 10993-5 [14]. The extracts were prepared as described in Section 2.4. The undiluted extracts of the alloys were toxic to both tested cell lines and almost completely inhibited the metabolic activity of both cell lines (relative metabolic activity was 7% or less; data not shown). Therefore, only results with 50% extracts are presented. The Zn and Ag concentrations in the extracts were measured using ICP-MS. The concentration of Ag in the diluted extracts was below 1.8 nmol∙L^−1^ and was considered to have a minor effect. The concentration of Zn, ranging between 72 and 102 µmol∙L^−1^, is shown for particular alloys on the labels above the columns of the graph. We did not observe any increase in pH after extraction using phenol red in the cultivation medium as an indicator. This is in accordance with our previous studies [40].

Figure 5 shows that the extracts were more toxic to the L929 cell line. For example, the same extract of Zn-1Mg in DMEM/Ham’s F-12 medium (Zn concentration = 73 µmol∙L^−1^) did not have any adverse effect on the hFOB 1.19 cell line; however, the metabolic activity of L929 was decreased to 2%.

Generally, the extracts prepared in MEM were more toxic for the hFOB 1.19 cell line compared to the extract prepared in DMEM/Ham’s F-12, even when they contained less Zn. For example, the Zn-1Mg-1Ag extracts in DMEM/Ham’s F-12 medium (Zn concentration = 90 µmol∙L^−1^) did not decrease the metabolic activity of the hFOB 1.19 cells, while the Zn-1Mg-1Ag extract in MEM (Zn concentration = 72 µmol∙L^−1^) decreased the metabolic activity of the hFOB 1.19 cells below 20%.

The metabolic activity of L929 after exposure to all of the extracts was 2% or less. Therefore, the difference in toxicity of the extract in MEM and DMEM/Ham’s F-12 is less evident.

## 4. Discussion

The human fetal osteoblast cell line hFOB 1.19 is a promising model for the *in vitro* testing of biomaterials due to its similarity to primary osteoblasts and its unlimited supply caused by conditional immortalization. However, the discrepancies in the cultivation protocols raise the question of what conditions should be chosen during the biomaterial tests.

First, we compared the sensitivity of hFOB 1.19 to Zn^2+^ in the presence and absence of the selection reagent, G418. When establishing this cell line by Harris et al. [20], G418 was used for the selection of transfectant and for additional maintenance during cultivation, since the desirable phenotype can be lost with the withdrawal of selective pressure [45]. Here, we have shown that the sensitivity of hFOB 1.19 to zinc is strongly affected by the presence of G418 both after short- and long-term incubation. We observed a higher sensitivity of hFOB1.19 to zinc in the absence of G418 (Figure 2). Some authors tend to use G418 not only during cultivation, but also during biomaterial tests [24,32,33], or the usage of G418 is not specified ([31,37]). In our opinion, in the case where the hFOB 1. 19 cell line is used as an appropriate cell model (mature osteoblastic phenotype is preferred, immortalization is not necessary), the G418 selection reagent does not need to be used during the experiment.

Furthermore, we studied the sensitivity hFOB 1.19 to zinc at different temperatures. We observed a higher sensitivity of hFOB 1.19 to Zn^2+^ at a restrictive temperature. Harris compared proliferation at 33.5 °C (as permissive), 38, and 39 °C (both considered restrictive). The doubling time of hFOB 1.19 at 34 °C was ~36 h, and at 38 °C, it was >96 h [20]. However, there is no clear consensus on whether the temperature of 37 °C should be considered permissive or restrictive. The higher metabolic activity and lower sensitivity to ZnCl_2_ of hFOB 1.19 cells at a permissive temperature (34 °C) in our case was probably due to their ongoing proliferation. The resulting cell density was higher and, therefore, the concentration of Zn^2+^ per cell was lower. It is probably also due to the fact that at a temperature of 39.5 °C, the cells are stressed and thus more sensitive. This is supported by the fact that we observed an elevated level of heat shock proteins (HSPs) in hFOB 1.19 at 39.5 °C (our unpublished results, [46]). In our opinion, the physiological temperature of 37 °C is ideal for the testing of biomaterials using the hFOB 1.19 cell line. HSPs are probably not overexpressed and the temperature is sufficient for the development of the mature phenotype [20].

The comparison of the sensitivity of hFOB 1.19 to Zn^2+^ with the results obtained by other authors is difficult, since other studies using hFOB 1.19 to test degradable biomaterials with Zn (Table 1) do not specify the concentration of Zn^2+^ released to the medium in contact with the cells.

We also compared the sensitivity of the hFOB 1.19 and L929 cell lines to Zn^2+^ (Figure 3). The higher sensitivity of L929 to Zn^2+^ compared to hFOB 1.19 is in agreement with our previous results showing that another osteoblast-like cell line, U-2 OS, also withstands a higher concentration of Zn^2+^ (200 µmol∙L^−1^, when tested in MEM medium) compared to the L929 fibroblast-derived cell line [19]. We did not compare hFOB 1.19 and U-2 OS directly, but from testing under the same conditions, we can presume that hFOB 1.19 cell line is more sensitive than U-2 OS. Cell lines derived from bone cells (whether primary or cancerous) appear to be less sensitive to Zn^2+^ than the commonly used L929 line, and, therefore, their use in the testing of biomaterials intended for orthopedic implants may play a role in bringing the *in vitro* system closer to the real situation in the body.

We have also found that the type of medium plays a prominent role. Our previous results showed that ZnCl_2_ solutions were less toxic to L929 and U-2 OS in DMEM (Sigma, D0819) than in MEM (Sigma, M0446) [19]. This was confirmed for hFOB 1.19 (Figure 4), where the toxic effect of ZnCl_2_ was lower in the case of rich DMEM/Ham’s F-12 compared to the minimal medium MEM. Surprisingly, ZnCl_2_ was less toxic to L929 in MEM than in DMEM/Ham’s F-12 (Figure 4). DMEM/Ham’s F-12 is a mixture of component-rich Ham’s F12 medium and the nutrient-rich DMEM medium. However, some of its components (e.g., ferrous sulphate, HEPES) may have negative effects on certain cell types [47]. It was also possible that the L929 cells were unable to adapt to a new medium. Also, the availability of Zn in media can differ. For example, riboflavin (0.27 μmol∙L^−1^ in MEM vs. 1.0 μmol∙L^−1^ in DMEM/Ham’s F-12) forms toxic complexes with certain metals [48].

We also prepared extracts of Zn-based biomaterials and tested the cytotoxicity of these extracts on the two cell types. The materials selected in the presented study are considered promising for future applications in the development of medical devices. The chemical composition of these selected materials, especially its Mg content, is beneficial for improving mechanical strength and maintaining an adequate degradation rate of the materials [4,40,41,49]. The suggested processing by powder metallurgy (PM), including the preparation of alloy powder by mechanical alloying and its compaction by extrusion at 200 °C and an extrusion ratio equal to 25, enables the formation of extremely fine-grained homogeneous microstructures, leading to superior mechanical properties and uniform degradation [42,43,50,51]. Silver is used particularly to improve materials’ plasticity, but also the antibacterial and anti-inflammatory properties [52]. During the degradation process of alloy, silver is released into the surrounding tissue, where it fulfils the antibacterial effect and subsequently is gradually eliminated from the human body by the liver and kidneys [53].

Extracts of the aforementioned materials were more toxic for L929 than for hFOB 1.19 (Figure 5). This is in agreement with the results with ZnCl_2_ solutions (Figure 3). We prepared the extracts of the materials in both types of media. Generally, the extracts prepared in MEM were more toxic for the hFOB 1.19 cell line compared to the extract prepared in DMEM/Ham’s F-12, even when they contained less Zn (Figure 5). Differences in the media’s composition probably improved the cell fitness and decreased the cytotoxic effect of extracts in DMEM/Ham’s F-12 (rich medium) compared to the MEM medium (minimal medium), similarly to our previous study with ZnCl_2_ solutions in MEM and DMEM [19].

## 5. Conclusions

The human fetal osteoblast cell line hFOB 1.19 is a promising model for the *in vitro* testing of degradable zinc-based biomaterials intended for orthopedic applications due to its resemblance to primary osteoblasts and to its unlimited supply caused by conditional immortalization. We have shown that hFOB 1.19 cell line was less sensitive to Zn^2+^ than L929, recommended by the ISO-10993 standard [14]. This standard was originally developed for non-degradable materials and, therefore, in case of degradable materials, often leads to exaggerated, i.e., more toxic, responses than expected *in vivo*. Thus, less sensitive cell lines and, at the same time, cells resembling osteoblasts seem to be a more convenient model for the *in vitro* testing of degradable materials for orthopedic applications. We have also shown that the cultivation conditions of hFOB 1.19 during testing affects the results: most importantly, (1) the presence of G418 used as a selection reagent decreased the sensitivity of hFOB 1.19 to Zn^2+^, and (2) hFOB 1.19 were more sensitive to Zn^2+^ at elevated (restrictive) temperatures. The cultivation conditions should therefore be chosen with caution.

## Figures and Tables

**Figure 1 materials-17-00915-f001:**
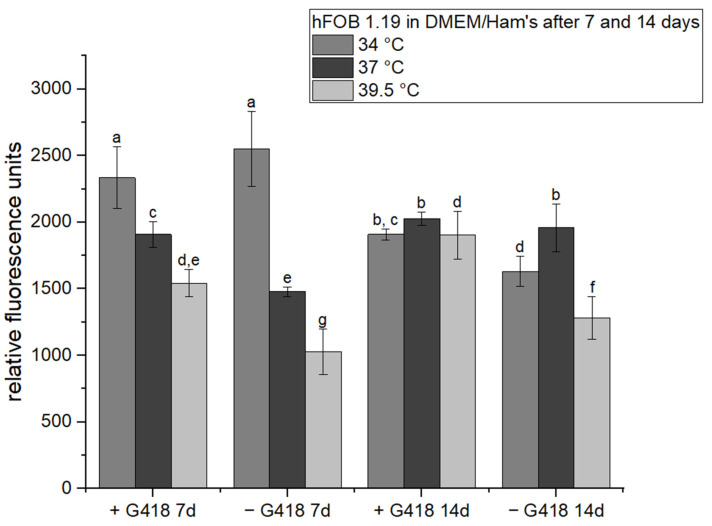
Resorufin fluorescence produced by hFOB 1.19 cells after 7 and 14 days of cultivation in the presence (+) and absence (−) of the G418 selection agent. The error bars indicate the sample standard deviation of three measurements (three wells). Differences (*p* < 0.05) among groups are indicated by letters (one-way ANOVA followed by Tukey’s test).

**Figure 2 materials-17-00915-f002:**
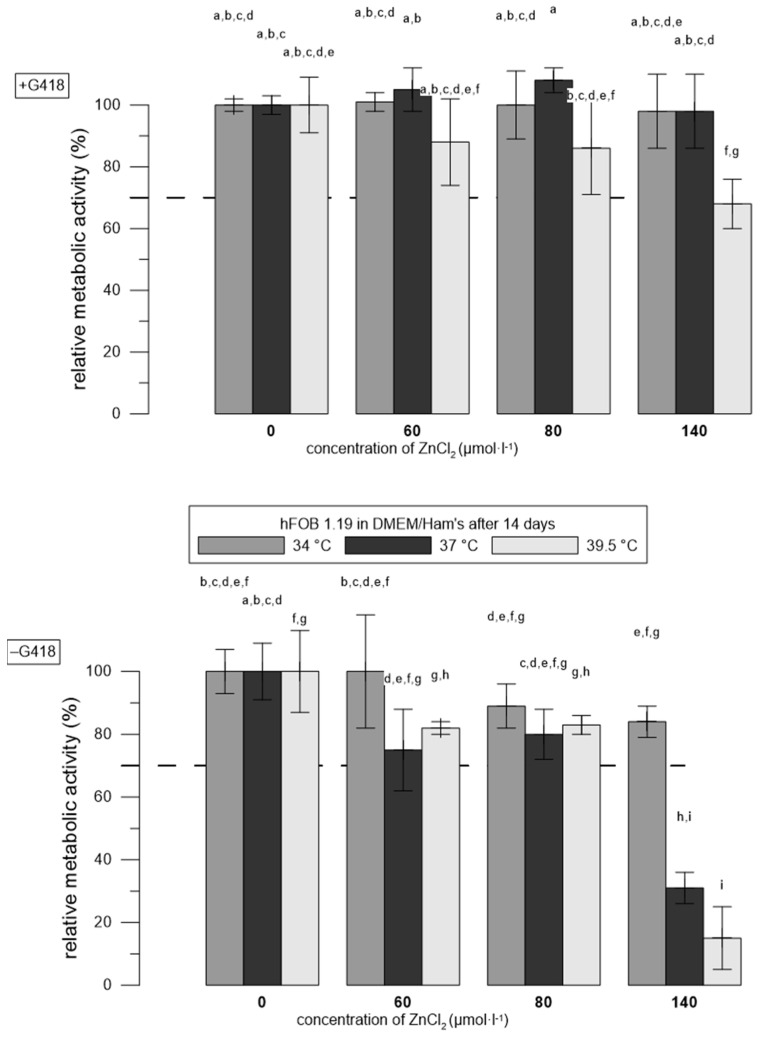
Relative metabolic activity of hFOB 1.19 cells after 14 days of differentiation in the presence (+) and absence (−) of the selection agent, G418, after 1 day of incubation with ZnCl_2_ solutions. Metabolic activity is expressed as a percentage (the negative control of 0 μmol∙L^−1^ of ZnCl_2_ represents 100%). Error bars indicate the sample standard deviation of three measurements (three wells). Differences (*p* < 0.05) among groups calculated from fluorescence values are indicated by letters (one-way ANOVA followed by Tukey’s test).

**Figure 3 materials-17-00915-f003:**
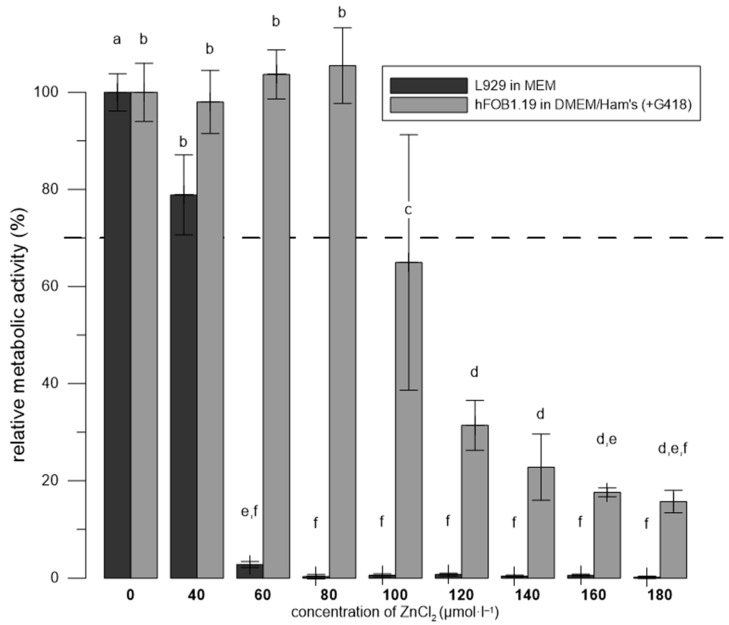
Relative metabolic activity of hFOB 1.19 and L929 cells after 1 day of incubation with ZnCl_2_ solutions. Metabolic activity is expressed as a percentage (negative control lacking ZnCl_2_ represents 100%). The error bars indicate the sample standard deviation of six measurements (six wells). Differences (*p* < 0.05) among groups calculated from fluorescence values are indicated by letters (one-way ANOVA followed by Tukey’s test).

**Figure 4 materials-17-00915-f004:**
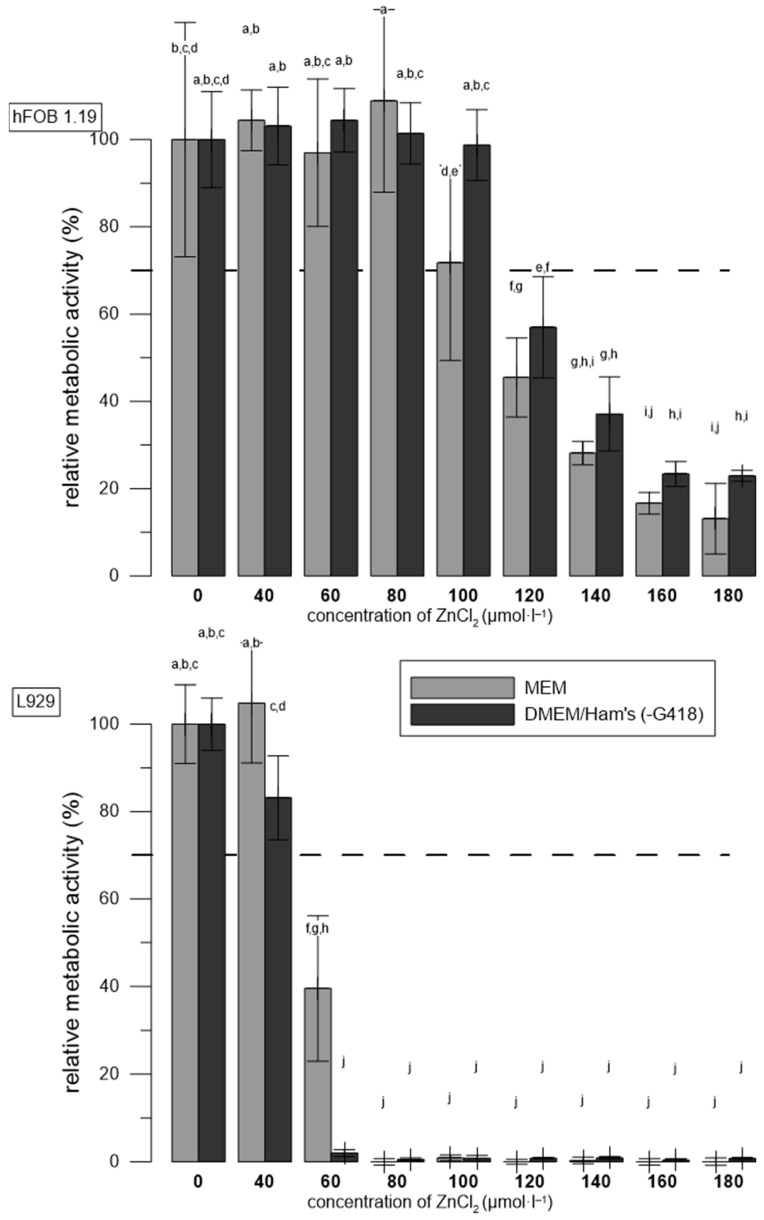
Relative metabolic activity of hFOB 1.19 and L929 cells after 1 day of incubation with ZnCl_2_ solutions. Metabolic activity is expressed as percent (negative control without ZnCl_2_ represents 100%). Error bars indicate the sample standard deviation of six measurements (six wells). Differences (*p* < 0.05) among groups are indicated by letters (one-way ANOVA followed by Tukey’s test).

**Figure 5 materials-17-00915-f005:**
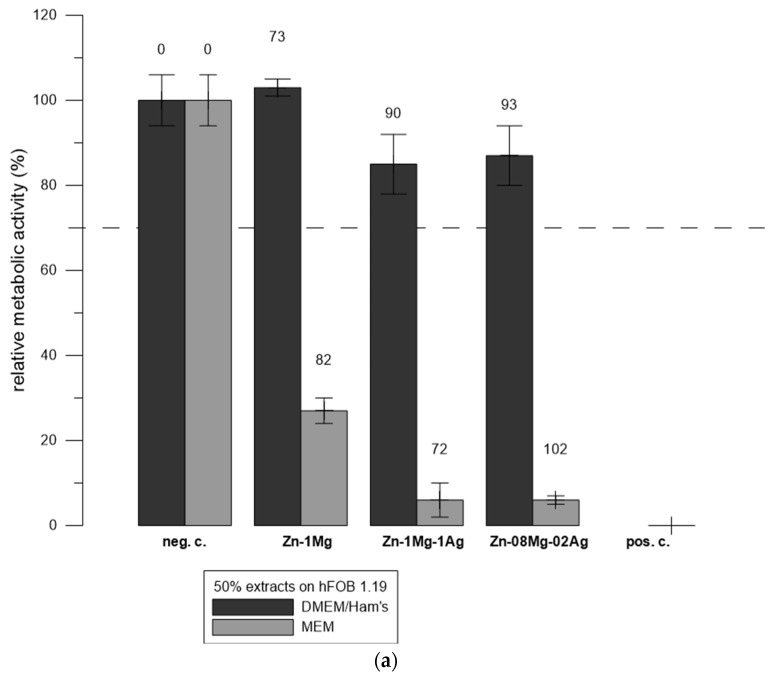
Relative metabolic activity of hFOB 1.19 (**a**) and L929 (**b**) cells after 1 day of incubation with extracts of Zn-based materials. Metabolic activity is expressed as a percentage (negative control of sole medium—represents 100%). Error bars indicate the sample standard deviation of six measurements (six wells). The labels above the columns indicate the concentration of Zn in µmol∙L^−1^ measured by ICP-MS.

**Table 1 materials-17-00915-t001:** Studies using hFOB 1.19 for the testing of degradable biomaterials containing Zn (HA = hydroxyapatite).

Material	G418	Temperatures	References
sol–gel-prepared glass materials with ZnO	not mentioned	37 °C as permissive 39.5 °C as restrictive	[29]
Zn-doped HA coatings	not mentioned	37 °C	[34]
Reactive interfaces based on hydroxides-Zn rich HA	present	37 °C	[35]
3D porous granules based on Zn-containing CaPs	present	34 °C	[36]
Zn-doped HA nanopowders	not mentioned	34 °C	[37].
HA and bimetallic nanocomposite of ZnO–Ag	present	34 °C	[38]
Zn-doped nanoHA-based bone scaffolds	present	34 °C	[39]

**Table 2 materials-17-00915-t002:** Materials used for testing.

Designation (Composition in wt.%)	Synthesis	Processing Conditions (Temperature and Extrusion Ratio)
Zn-1Mg	Powder metallurgy	Extrusion of powder billets at 200 °C and extrusion ratio of 25
Zn-1Mg-1Ag	Powder metallurgy	Extrusion of powder billets at 200 °C and extrusion ratio of 25
Zn-0.8Mg-0.2Ag	Conventional casting and extrusion	Extrusion of casted ingot at 200 °C and extrusion ratio of 25

## Data Availability

The datasets generated during and/or analyzed during the current study are available from the corresponding author on reasonable request. Data from ICP-MS measurements are accessible via the Zenodo repository: https://doi.org/10.5281/zenodo.10589660 (accessed on 7 February 2024).

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
