# Peer review of "Characterization of hFOB 1.19 Cell Line for Studying Zn-Based Degradable Metallic Biomaterials"

_materials, 2024, doi:10.3390/ma17040915_

Round 1
Reviewer 1 Report
Comments and Suggestions for Authors
In this manuscript, authors mainly studied the sensitivity of hFBO1.19 cell line to Zn-based degradable metallic biomaterials. Two mediums with and without G418 were used to observe the role of G418 for hFBO1.19 and L929 cells. Three extracts from Zn-1Mg, Zn-1Mg-1Ag, and Zn-0.8Mg-0.2Ag were used to demonstrate the cytotoxicity of hFBO1.19 and L929. Most experimental results supported the conclusion. Therefore, some issues need to be addressed. Some questions and comments are as below.
1. Please correct the subscript and superscript of chemical formula.
2. Why didn't cells directly use Zn-base biomaterials to incubation? In this study, authors used the extracts.
3. If Zn-base biomaterials are heated at 37 °C, what amount of Zn2+ were released from materials? This condition seems to close the actual status.
4. Why was higher relative metabolic activity of hFOB 1.19 in ZnCl2 (Zn ion: 60~100 μmol∙l-1) than that in Zn-1Mg (Zn ion: 82 μmol∙l-1), Zn-1Mg-1Ag (Zn ion: 72 μmol∙l-1), or Zn-0.8Mg-0.2Ag (Zn ion: 102 μmol∙l-1)? This indicates that other factors exist. What are factors?
5. Some standard deviation in all figures are very unnatural, even some S.D. are not straight. Should they have the same presentation?
6. In discussion section, why did authors show the first paragraph?
7. Conclusion is too simple.
Comments on the Quality of English LanguageEnglish Language is well.
Author Response
Reviewer 1
- Please correct the subscript and superscript of chemical formula.
Answer 1: We thank you for careful reading, all the subscript and superscript errors were corrected.
- Why didn't cells directly use Zn-base biomaterials to incubation? In this study, authors used the extracts.
Answer 2: Thank you for your comment. The test with extract is a standardised procedure described in ISO standard (10993-5). Direct incubation of cells with biomaterials is another possibility, however, in case of degradable metallic biomaterials usually leads to overestimated toxicity and therefore, this method was not performed in this case.
- If Zn-base biomaterials are heated at 37 °C, what amount of Zn2+were released from materials? This condition seems to close the actual status.
Answer 3: Thank you for your comment. The materials were incubated at 37 °C in the medium (as described in the section 2.6) and the amount of released Zn2+ is in the Fig. 5 as number above columns as described in the caption of the Fig. 5.
- Why was higher relative metabolic activity of hFOB 1.19 in ZnCl2(Zn ion: 60~100 μmol∙l-1) than that in Zn-1Mg (Zn ion: 82 μmol∙l-1), Zn-1Mg-1Ag (Zn ion: 72 μmol∙l-1), or Zn-0.8Mg-0.2Ag (Zn ion: 102 μmol∙l-1)? This indicates that other factors exist. What are factors?
Answer 4: Thank you for your comment. In case of alloys with Ag, there could be the slight influence of synergic action of silver ions, despite the fact that the measured concentration of Ag was very low. In case of Zn-1Mg alloy, we were not able to identify any factor. We were trying to cover all possible factors (e.g. monitoring of pH, which was not different from the control), however the process of corrosion brings many other factors to play, that can’t be easily evaluated. Traces of impurities from the process of preparation could possibly also play a synergic role.
- Some standard deviation in all figures are very unnatural, even some S.D. are not straight. Should they have the same presentation?
Answer 5: Thank you for noticing it. We inserted a new version of the figure 5a.
- In discussion section, why did authors show the first paragraph?
Answer 6: We apologize for the mistake. It is just a residue from the template left there accidentally. We have removed the text.
- Conclusion is too simple.
Answer 7: Thank you for your comment. We extended the conclusion.
Reviewer 2 Report
Comments and Suggestions for Authors
Authors presented a standard paper about the sensibility of cell lines to Zn biomaterials. It offers some interesting results about the response of two different cell lines to the materials under different conditions, but I think that there is not enough novelty to be published. The paper can be completed with proteomic analysis (specially those that can be affect by Zn exposure) in order to provide a more detailed approximation to this phenomenon.
Other suggestions that authors could take into account.
- Add references in Table 2 for each case (references 40 to 43)
- Add the statistical analysis procedure in materials and methods section (ANOVA and Tukey)
- Remove the first paragraph from Discussion section. It is copied from the template.
- Conclusions should be longer. One paragraph is not enough to summarize the main conclusions of the study.
Author Response
Reviewer 2
Authors presented a standard paper about the sensibility of cell lines to Zn biomaterials. It offers some interesting results about the response of two different cell lines to the materials under different conditions, but I think that there is not enough novelty to be published. The paper can be completed with proteomic analysis (specially those that can be affect by Zn exposure) in order to provide a more detailed approximation to this phenomenon.
Thank you for the comment. Because there is growing number of works using hFOB 1.19 cell lines, we wanted to point out relatively fast to the problems connected with using them. We aim to continue with more detailed analysis, including proteomics, in the future, unfortunately it can’t be done in time given for the revisions.
1) Add references in Table 2 for each case (references 40 to 43)
Answer 1: Thank you for noticing. For clarity, we eventually decided to describe the preparation of the materials in more details directly in section 2.3, because the cited articles do not correspond to the exact types of alloys.
2) Add the statistical analysis procedure in materials and methods section (ANOVA and Tukey)
Answer 2: The statistical analysis procedure was added to “materials and methods” section.
3) Remove the first paragraph from Discussion section. It is copied from the template.
Answer 3: We apologize for the mistake, the paragraph was removed.
4) Conclusions should be longer. One paragraph is not enough to summarize the main conclusions of the study.
Answer 4: Thank you for your comment. We extended the conclusion.
Reviewer 3 Report
Comments and Suggestions for Authors
The author in the manuscript titled “Characterization of hFOB 1.19 cell line for studying Zn-based degradable metallic biomaterials” has investigated and demonstrated the cultivation condition of hFOB 1.19 during biomaterial testing. I recommend the publication of the manuscript after minor revision.
1) Page 10, first paragraph of Discussion instruction should be removed.
2) The author states in Page11, line 319-320 that the cell lines derived from bone cells are less sensitive, is there a plausible reason for the sensitivity to be less.
3) The author talks about the mechanical strength, is there any literature report which illustrates the improvement in such properties along with cytotoxicity test.
4) Page 12, line 348 the toxicity for L929 is more than hFOB 1.19, is there any plausible reason for this? Is there any characteristic structural difference which results in this? Based on the difference can a set of family be categorized based on similarity.
Author Response
Reviewer 3
- Page 10, first paragraph of Discussion instruction should be removed.
Answer 1: We apologize for the mistake, the paragraph was removed.
- The author states in Page11, line 319-320 that the cell lines derived from bone cells are less sensitive, is there a plausible reason for the sensitivity to be less.
Answer 2: Thank you for the comment. L929 cell line is generally considered to be sensitive to external factors and therefore, its usage for biological testing is recommended by ISO standard. However, the standard was originally developed for non-degradable materials and therefore, in case of degradable materials often leads to more toxic response than expected in vivo. Less sensitive cell lines and, at the same time, cells resembling osteoblasts seem to be more convenient model for in vitro testing of degradable materials for orthopaedic applications.
- The author talks about the mechanical strength, is there any literature report which illustrates the improvement in such properties along with cytotoxicity test.
Answer 3: According to our yet unpublished results, the processing of alloys by powder metallurgy significantly affected mechanical properties and led to the extraordinary values of TYS and UTS highly exceeding other zinc-based materials. Some concomitant decrease in elongation is observed although the values are still sufficient for wide range of applications in medicine. However, this manuscript is dealing mainly with effect of media and conditions used for testing. For comparison of cytocompatibility of alloys prepared by different technologies, more replicates and different experimental setup would be needed.
- Page 12, line 348 the toxicity for L929 is more than hFOB 1.19, is there any plausible reason for this? Is there any characteristic structural difference which results in this? Based on the difference can a set of family be categorized based on similarity.
Answer 4: L929 is generally very sensitive, which is probably the reason why they are selected by ISO standard (also answer 2). The differences between the cell lines could be dramatic and unification should be considered, on the other hand, there cannot be one cell line recommended for all the tests because the enormous diversity in the conditions used in various experiments. To our knowledge there is no attempt to categorize the cell lines according to their sensitivity, the main reason is probably generally low pressure by the authorities to force use of various cell lines during the testing.
Round 2
Reviewer 1 Report
Comments and Suggestions for Authors
Reviewer 2 Report
Comments and Suggestions for Authors
Authors have adressed some of my comments, but the novelty of the work is still questioned. More relevant experiments are needed to prove the novelty of the work.